# CMOS Low-Power Optical Transceiver for Short Reach

**DOI:** 10.3390/mi16050587

**Published:** 2025-05-17

**Authors:** Ruixuan Yang, Yiming Dang, Jinhao Chen, Dan Li, Francesco Svelto

**Affiliations:** 1Faculty of Electronic and Information Engineering, Xi’an Jiaotong University, Xi’an 710049, China; yrxay358@stu.xjtu.edu.cn (R.Y.); dym789521@stu.xjtu.edu.cn (Y.D.); jhchen22@stu.xjtu.edu.cn (J.C.); 2Department of Electrical Computer and Biomedical Engineering, University of Pavia, 27100 Pavia, Italy

**Keywords:** low power, optical transceiver, CMOS, driver, high-impedance anode driving, VCSEL, TIA, VGA

## Abstract

The emergence of the AI era driven by Large Language Models (LLMs) and the next-generation high-definition multimedia interface for immersive technologies (AR/VR/metaverse) have created an unprecedented demand for high-bandwidth interconnects. While optical communication systems provide a broad bandwidth, their relatively low power efficiency continues to limit their deployment in new applications. This work addresses the power efficiency challenges in CMOS optical transceiver design, leveraging the inherent cost and integration advantages of CMOS technology. After outlining the design principles for low-power optical transmitter (Tx) and receiver (Rx) design, we present a comprehensive design of a low-power optical transceiver chipset implemented in 28 nm CMOS. The Tx features a high-impedance asymmetric current-steering output stage with a stacked architecture that facilitates unipolar power supply operation for the efficient anode driving of a common-cathode VCSEL array and achieved a power efficiency of 1.59 pJ/bit. The Rx incorporates a tail-current-controlled Cherry–Hooper-based variable gain amplifier (VGA), which achieved a transimpedance gain that ranged from 68.4 to 78.5 dBΩ and a power efficiency of 1.06 pJ/bit. The Rx–Tx back-to-back measurements confirmed successful data transmission at 4 × 20 Gbps, which demonstrated an overall power efficiency of 2.65 pJ/bit.

## 1. Introduction

The rapid expansion of emerging industries, such as big data, artificial intelligence-generated content (AIGC), and virtual and augmented reality (VR/AR), is driving an ever-increasing demand for computing power and data transmission density, thereby placing greater requirements on high-bandwidth interconnects [1]. Although optical communication systems provide broad bandwidths, their relatively low power efficiency (>10 pJ/bit [2]) continues to limit their deployment in new applications, such as module-to-module and chip-to-chip communications.

As shown in Table 1, different scenarios put forward different requirements and expected power consumption ranges for optical interconnection. Short-reach optical interconnects, which are designed for distances under 100 m, such as intra-rack or server-to-switch communication, are tailored for high-performance computing (HPC) units’ inner interconnects for data centers. These systems require ultra-low latency below 10 nanoseconds and a high bandwidth density higher than $1\;\mathrm{Tbps}/{\mathrm{mm}}^{2}$ [3,4,5,6]. Especially in AI scale-up networks, their strict latency requirements limit the usage of a retime structure with a clock data recovery circuit, thus limiting higher than a 50 GBd data transmission. Key enabling technologies include vertical-cavity surface-emitting lasers (VCSELs) [7], silicon photonics (SiPh) [8], and multi-mode fiber (MMF) [9], achieving power efficiencies of 1–5 pJ/bit [10,11] to support dense, energy-efficient integration. Medium-reach optical interconnects, spanning distances from 100 m to 10 km for applications like inter-rack connectivity and campus networks, address distributed training and edge-to-cloud coordination [12]. These systems emphasize moderate latency between 10 and 100 nanoseconds, scalability for multi-node AI scale-out networks, and cost-effective mid-scale deployments. By leveraging single-mode fiber (SMF), electro-absorption modulated lasers (EMLs), and silicon photonics, they strike a balance between reach and energy consumption, operating at 5–15 pJ/bit. Long-reach optical interconnects operating over distances exceeding 10 km, such as metro/core networks and cross-continental links, facilitate distributed AI model synchronization and cloud-to-edge data pipelines. Critical demands include robust reliability, support for coherent transmission schemes, and resilience to chromatic dispersion and signal attenuation. Employing coherent optics, erbium-doped fiber amplifiers (EDFAs), and wavelength-division multiplexing (WDM), these systems achieve extended reach at the cost of higher power consumption (15–100+ pJ/bit) due to advanced DSP processing and optical amplification requirements [13,14,15].

Due to the inherent limitation of heat dissipation caused by space limits, the short-reach optical interconnect is more sensitive to power consumption. Consequently, power-optimized optical transceivers become essential to meet the dual requirements of low latency and high bandwidth density in short-reach interconnects for AI/ML and HPC applications. Despite its ability to significantly improve the bit error rate (BER) performance [16,17], the on-chip clock data recovery (CDR) circuit suffers from substantial power consumption (3.7 pJ/bit for Rx CDR in [17]), which ultimately compromises its overall power efficiency. Non-retimed transceivers provide a viable solution for short-range transmission scenarios with low insertion losses [18,19] for a better power efficiency (∼1.8 pJ/bit for Tx and ∼1.2 pJ/bit for Rx). However, further power optimization can be realized through strategic circuit modifications, such as deploying a high-impedance asymmetric current-steering output stage with a stacked architecture for the Tx and integrating a tail-current-controlled Cherry–Hooper-based VGA in the Rx.

This work presents a comprehensive design methodology for low-power optical transceivers targeting short-reach applications, achieving more than 10% power reduction for both Tx and Rx compared with the previous state of the art [18,19]. Section 2 details the architectural and system-level design considerations, while Section 3 provides the complete circuit implementation with key optimization techniques. The fabricated prototype’s measurement results are shown in Section 4 and compared with other state-of-the-art designs. Finally, Section 5 concludes the transceiver performance and discusses potential applications.

## 2. Low-Power Design Methodology for Optical Transceivers

Low-power optical interconnects implementation requires a holistic co-design approach spanning the system architecture, device characteristics, and circuit implementation. Detailed system-level consideration and circuit-level techniques for low-power optical transceiver design are shown in the following sections.

### 2.1. System-Level Consideration

At the system level, the first consideration is the optical module structure. As shown in Table 2, optical modules are gradually developing toward miniaturization, high integration, and low power consumption, especially for short-reach applications, ranging from conventional pluggable modules to co-packaged optical modules (CPOs) [6], with their system block diagrams shown in Figure 1. The main strategies used to reduce power consumption are as follows:Shorten the electrical interface: change the optical module structure from pluggable optical modules with long PCB traces to CPOs with ultra-short interconnects, reducing the electrical insertion losses and enabling low-complexity equalization (e.g., CTLE, FFE, DFE) integration with the analog front end.Eliminate high-order DSP equalization: short-reach application relaxes the dispersion tolerance, eliminating the need for DSP for dispersion compensation.Retimed Tx/Rx topology: integration with clock data recovery (CDR) circuits brings better signal integrity with a lower jitter, which eliminates power-hungry, high-speed analog-to-digital converters (ADCs) at the receiver.

The second consideration is the selection of modulators. As mentioned above, different scenarios put forward different requirements for the laser and fiber selection due to variable link losses and margins of chromatic dispersion. The characteristics of common modulators are shown in Table 3. Although external modulation exhibits a reduced chirp effect, making it suitable for long-range, high-speed communication, its implementation necessitates additional lasers and results in higher power consumption. Consequently, direct modulation is generally preferred in low-power designs. In comparison with high-power distributed feedback lasers (DFBs), vertical-cavity surface-emitting lasers (VCSELs) offer advantages such as lower manufacturing costs and reduced driving currents, making them a more attractive option for low-power short-reach systems.

And for the EICs, advanced CMOS nodes (28 nm and below) provide dual advantages: lower dynamic power consumption through VDD scaling (0.5–0.9 V core voltages) and reduced leakage currents via FinFET/nanosheet architectures [20], and optimized device sizing in critical paths—particularly for the VCSEL driver and TIA front-end—enhances the $fCV^{2}$ dynamic power efficiency while maintaining an optimal optical modulation performance.

### 2.2. Circuit-Level Techniques

Circuit-level techniques further complement these foundational advancements in low-power design.

#### 2.2.1. Low-Power Techniques for Drivers

The design of low-power drivers mainly focuses on the output stage power consumption reduction, and its main methods include the following:Open-drain output topology: employing compact uncooled modulators with direct coupling avoids transmission line effects, eliminating the need for termination for impedance matching, as shown in Figure 2b, where the open-drain output topology can be utilized to eliminate the shunt current of the terminal resistors and improve the current modulation efficiency.Active back termination: co-designing driver output impedance with VCSEL parasitics avoids reflections using active back termination (ABT) [21] to improve the modulation efficiency, as shown in Figure 3.Stack structure: For direct-coupled VCSEL anode driving, a higher common-mode voltage is needed at the output node. To eliminate the need for additional low-dropout regulators (LDOs), the output stage can be stacked on top of the previous stage. This not only raises the output node voltage level but also reuses the current, thereby achieving significant power savings [16].

#### 2.2.2. Low-Power Techniques for TIAs

Single-ended front-end TIA without a dummy TIA: Though integration with a dummy TIA can reduce the offset of single-ended to differential converter (S2D), as shown in Figure 4a, it brings additional power consumption. By balancing the parasitic capacitance and area consumption, AC coupling can be utilized to eliminate the offset without additional power consumption.Adaptive biasing: A received signal strength indicator (RSSI) dynamically adjusts the TIA gain with a variable power distribution to match the received optical power. This avoids over-biasing under high-signal conditions. And noise-adaptive biasing: lowering the bias currents in low-noise regimes (high input power) optimizes noise–power tradeoffs.Adaptive output swing: For the output buffer, which often utilizes the current-steering structure for impedance matching, the output swing is defined by the tail current, as shown in Figure 5. An adaptive output swing with a variable tail current can reduce the power consumption significantly when operating at a low output swing.

#### 2.2.3. Shared Low-Power Design Methods

And shared low power design methods for both drivers and TIAs include the following:On-chip inductors: by employing multiple bandwidth-boosting topologies with on-chip inductors, as detailed in [22], the bandwidth limitations of CMOS circuits can be effectively overcome, enabling a lower current consumption while maintaining the same bandwidth, as shown in Figure 6.Burst mode operation: burst mode operation shuts down the main path in the idle state, thus effectively reducing the power consumption.

This multi-layer co-design approach demonstrated a 2.65 pJ/bit aggregate efficiency in the transceiver prototype implementation, establishing this CMOS photoelectric as a viable path for terabit-scale, energy-efficient interconnects.

## 3. Circuit Design

Both the driver and the TIA comprise four channels. The driver operates with a single 3.3 V power supply, while the TIA requires a 0.9 V power supply for the main signal path and a 3.3 V power supply for photodiode (PD) biasing. Detailed designs of the driver and TIA are provided in the following sections.

### 3.1. Driver Design

The main signal path of the driver consists of a continuous-time linear equalizer (CTLE), which offers 1–6 dB of equalization at the Nyquist frequency; a two-stage pre-driver circuit (PDRV) that amplifies the signal to drive the output stage; and an AC-coupled output stage (MDRV), stacked atop the CTLE and PDRV, as shown in Figure 7, enabling current reuse for improved efficiency. The auxiliary circuits include a bandgap reference and biasing circuit (BG), a low-dropout (LDO) voltage regulator, and an I2C interface.

For high-density interface applications, direct DC-coupled anode driving of common-cathode VCSEL arrays is essential, as it enhances the interconnection density while minimizing the cost of peripheral components. In unipolar power supply designs, conventional cascaded driver architectures employ a high-voltage supply for the output stage to accommodate the elevated common-mode voltage requirement imposed by grounded-cathode VCSELs, as illustrated in Figure 8a. The DC level mismatch between the output stage and the previous stage leads to inevitable power waste. Although employing a negative supply enables the use of a low-voltage output stage [18], as depicted in Figure 8b, it introduces additional complexity and incurs higher costs in the power supply system. To address this, a stacked architecture that recycles the output stage current drastically reducing power consumption in high-voltage operation [16]. This approach also eliminates the need for a negative power supply while enabling direct DC-coupled anode driving of common-cathode VCSEL arrays.

The high-impedance asymmetric current-steering output stage is shown in Figure 9.

Unlike conventional resistive termination, the high-impedance output not only prevents voltage drops caused by tail current shunting in direct DC-coupled VCSELs—which would otherwise require a higher power supply for the output stage to maintain the DC operating point, leading to increased power consumption—but also enhances the modulation efficiency by ensuring the full injection of the modulation current into the VCSEL. Benefitting from the stacked structure combined with a high-impedance output stage, the driver achieved a power efficiency of 1.59 pJ/bit with an up to 6 mA bias current and an up to 6 ${\mathrm{mA}}_{\mathrm{pp}}$ modulation current of the VCSEL. Simulated eye diagrams of the driver with a variable modulation current and variable bias current for the VCSEL are shown in Figure 10. And the power breakdown when operated at $I_{\mathrm{mod}}$ = 6 mA and $I_{\mathrm{bias}}$ = 6 mA is shown in Figure 11.

### 3.2. TIA Design

As shown in Figure 12, the main signal path of the TIA comprises a front-end transimpedance amplifier (FE-TIA), which converts the current signal from the photodiode into a voltage signal; a single-ended-to-differential converter (S2D) that transforms the single-ended voltage signal from the FE-TIA into a differential signal; a main amplifier (MA) composed of three tail-current-controlled Cherry–Hooper-based variable gain amplifiers (VGAs), which further amplify the differential signal; and a feed-forward equalizer integrated buffer (FFE-BUF) that provides additional equalization. With this architecture, the TIA supports data rates of up to 20 Gbps.

The auxiliary circuits include an input DC current cancellation circuit (IDCC) that removes the DC current from the photodiode, ensuring the proper DC operating point of the FE-TIA, a DC offset cancellation loop (DCOC) that eliminates the output DC offset voltage, a received signal strength indicator circuit (RSSI) for measuring the intensity of the received optical signal, a signal loss detection circuit (LOS) to determine whether the optical signal is lost, a bandgap reference and biasing circuit (BG) for stable voltage and biasing current generation, a low-dropout voltage regulator (LDO) for power management, and an I2C interface for communication and control.

The FE-TIA was designed to achieve a closed-loop bandwidth exceeding 20 GHz while simultaneously meeting stringent low-power requirements and maintaining optimal noise performance. Given the power constraints, the TIA operates on a 0.9 V power supply for low power, circumventing the voltage margin loss associated with the use of low-dropout regulators (LDOs), which imposes stricter demands on the TIA’s noise performance. Although the three-stage inverter-based SF-TIA, as discussed in papers [23,24], offers a higher transimpedance gain and improved noise performance, its bandwidth limitations and high-order, high-frequency roll-off necessitate additional power consumption for high-order equalization circuits. To address these challenges, this work employed a single-stage inverter as the forward amplifier in the TIA, as shown in Figure 13a. Additionally, input series inductors ($L_{S}$) and bonding parasitic inductors ($L_{B}$) were utilized to enhance the FE-TIA’s bandwidth and further optimize its noise performance.

As shown in Figure 14, inverter-based voltage amplifiers can be categorized into three main types based on their load configuration: (a) Resistive load structure—the bandwidth of this configuration is constrained by the load resistance and output capacitance, making it challenging to achieve high voltage gain while maintaining a broad bandwidth. (b) $g_{m}/g_{m}$ structure—This design utilizes an inverter with its input and output shorted as the load, resulting in a relatively low gain. However, it is well-suited for applications that require high linearity. (c) Cherry–Hooper structure—This approach employs a shunt feedback transimpedance amplifier (SF-TIA) as the load, which provides a low impedance at both the input and output nodes of the TIA. As a result, it enables a high bandwidth while preserving a significant gain [25].

With the same load capacitances and identical 6 dB gain, the tree topologies achieved 3 dB bandwidths of 10.9 GHz, 10.6 GHz, and 19.9 GHz, thereby validating the aforementioned conclusion. Given the design objectives of low power consumption, high gain, and high bandwidth, the Cherry–Hooper structure was selected as the foundational architecture for the MA.

To minimize the eye diagram jitter caused by the nonlinear and limiting effects of the MA under large-signal conditions, the MA must incorporate a gain adjustment function. The schematic of the tail-current-controlled Cherry–Hooper-based variable gain amplifier (VGA) used in the MA is shown in Figure 15a.

The transconductance $g_{m}$ is regulated by adjusting the tail current, which, in turn, is controlled by varying the size of the tail resistance. This method not only simplifies the control circuit, as it only requires switching each stage on or off, but also reduces the power consumption when operating in low-gain mode. In low-gain mode, the switch LG is turned off, connecting $R_{S}$ and reducing the tail current and $g_{m}$, thereby reducing the gain. Conversely, in high-gain mode, the switch LG is activated, shorting $R_{S}$ and thereby increasing $g_{m}$. The MA, which is composed of three cascaded VGAs, achieved a gain adjustment range of 10 dB. With high gain, FFE enabled, and in the 200 mVppd output mode, the power breakdown is shown in Figure 16.

## 4. Measurement Results

Fabricated in a standard 28 nm CMOS process, the die areas of both the driver and TIA were 1.9 ${\mathrm{mm}}^{2}$. Both the driver and TIA were directly packaged on board (COB), as shown in Figure 17, and directly wire-bonded with 14 GHz bandwidth 850 nm multimode VCSELs and PIN-PDs, respectively. The measurement setup is shown in Figure 18. A 500 ${\mathrm{mV}}_{\mathrm{ppd}}$ PRBS-15 differential signal generated from a Keysight M8195A arbitrary waveform generator was fed into the driver.

As shown in Figure 19, the standalone driver test achieved data transmission rates of up to 25 Gbps. With a bias current of 6 mA and a modulation current of $6\;{\mathrm{mA}}_{\mathrm{pp}}$, the transmitter achieved an extinction ratio of 2.7 dB while maintaining a single-channel power consumption of 39.8 mW, which resulted in a power efficiency of 1.59 pJ/bit. The maximum extinction ratio of 3.1 dB was observed when the bias current was reduced to 4.5 mA while maintaining a modulation current of $6\;{\mathrm{mA}}_{\mathrm{pp}}$. Table 4 presents a comparison of the driver’s performance with previous works, highlighting its competitive efficiency.

Back-to-back (B2B) measurements were conducted to evaluate the performances of both the TIA and the optical link. An additional 7 dB attenuation was introduced using the Keysight N7768A variable optical attenuator, which corresponded to an input average optical power of −10 dBm to the TIA. Full-channel measurements were performed, with the resulting up to 20 Gbps eye diagrams shown in Figure 20 under the 200 ${\mathrm{mV}}_{\mathrm{ppd}}$ output mode.

And the output noise standard deviation was measured at the sampling oscilloscope without applying any input signal to the receiver. The total output noise standard deviation ($\sigma_{\mathrm{Total}}$) was $4.64\;{\mathrm{mV}}_{\mathrm{rms}}$, with its histogram shown in Figure 21. The TIA output noise was calculated from $\sigma_{\mathrm{TIA}}^{2}=\sigma_{\mathrm{Total}}^{2}-\sigma_{\mathrm{Scope}}^{2}$, where the noise standard deviation ($\sigma_{\mathrm{Scope}}$) of the disconnected sampling oscilloscope was measured to be $1.08\;{\mathrm{mV}}_{\mathrm{rms}}$. This translated to a TIA output noise of $4.30\;{\mathrm{mV}}_{\mathrm{rms}}$ and an input referred noise (IRN) of 1.64μA. Table 5 summarizes the performance of the TIA, showcasing its lower power consumption and reduced input-referred noise.

## 5. Conclusions

This work presents a highly energy-efficient 4 × 20 Gbps optical transceiver implemented in 28 nm CMOS, showcasing a key design approach for next-generation low-power optical interconnects. The transmitter leverages a high-impedance asymmetric current-steering output stage with a stacked architecture, enabling direct anode driving of a common-cathode VCSEL array while achieving an industry-leading power efficiency of <1.59 pJ/bit at data rates of up to 25 Gbps. On the receiver side, the TIA integrates a tail-current-controlled Cherry–Hooper-based VGA, delivering a wide tunable gain range of 68.4–78.5 dBΩ and a remarkable 1.06 pJ/bit efficiency at 20 Gbps operation. Successful back-to-back measurements at 20 Gbps per channel validate the design’s reliability, positioning this chipset as a compelling solution for emerging high-bandwidth, low-power optical links. Beyond its immediate application in data center interconnects, this work paves the way for scalable, CMOS-compatible optical I/O architectures that could address the escalating demands of AI-driven computing, 5G/6G infrastructure, and next-generation immersive systems.

## Figures and Tables

**Figure 1 micromachines-16-00587-f001:**
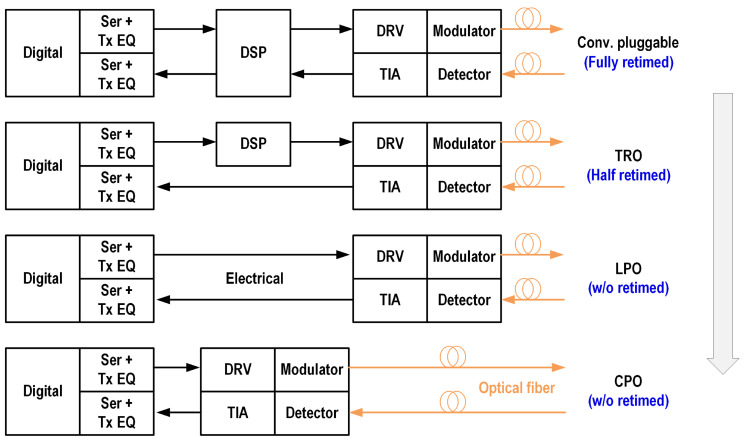
Block diagrams of the various structures of optical modules.

**Figure 2 micromachines-16-00587-f002:**
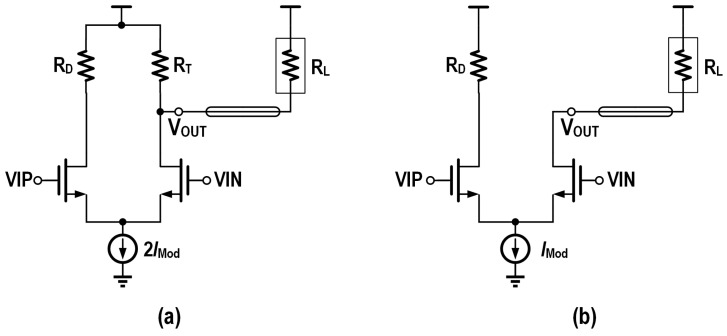
Schematic of current-steering output stage with (**a**) passive back termination and (**b**) open-drain topology.

**Figure 3 micromachines-16-00587-f003:**
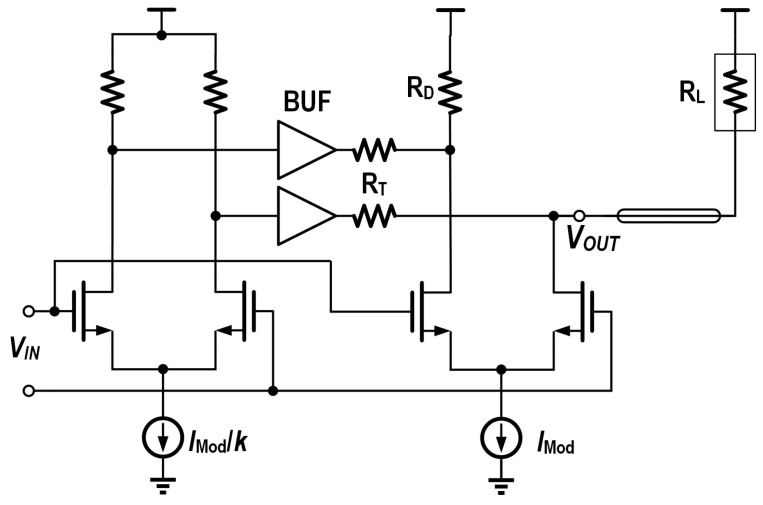
Schematic of the driver output stage with ABT.

**Figure 4 micromachines-16-00587-f004:**
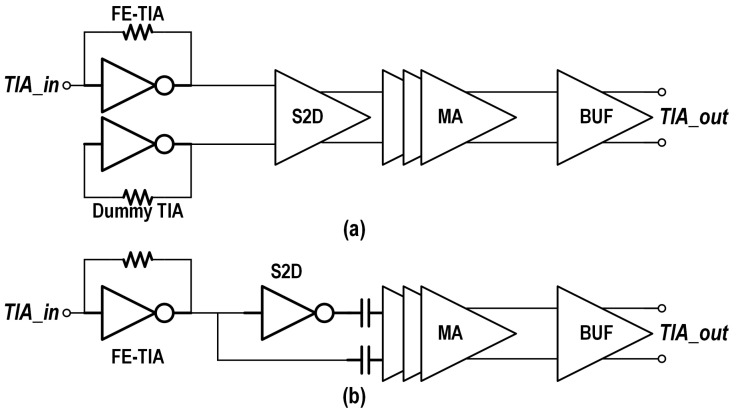
Schematic of the TIA (**a**) with and (**b**) without a dummy TIA.

**Figure 5 micromachines-16-00587-f005:**
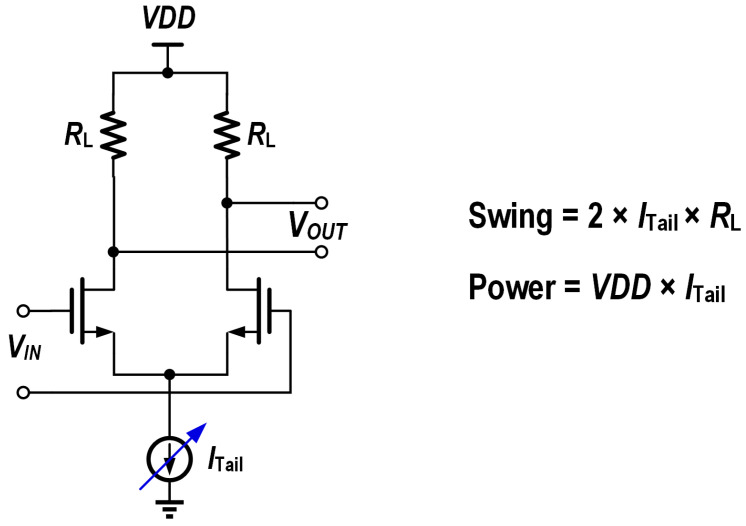
Schematic of current-steering output buffer with adaptive output swing.

**Figure 6 micromachines-16-00587-f006:**
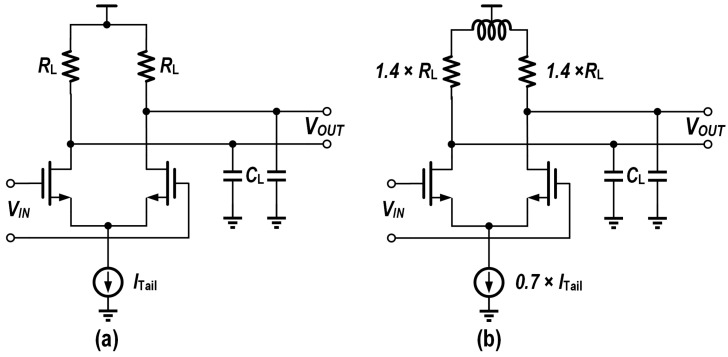
Schematic of current-steering stage with the same 3 dB bandwidth (**a**) without inductors and (**b**) with shunt peaking.

**Figure 7 micromachines-16-00587-f007:**
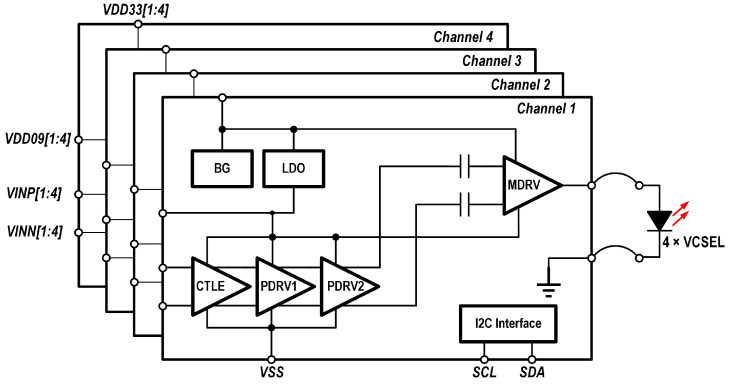
Block diagram of the driver.

**Figure 8 micromachines-16-00587-f008:**
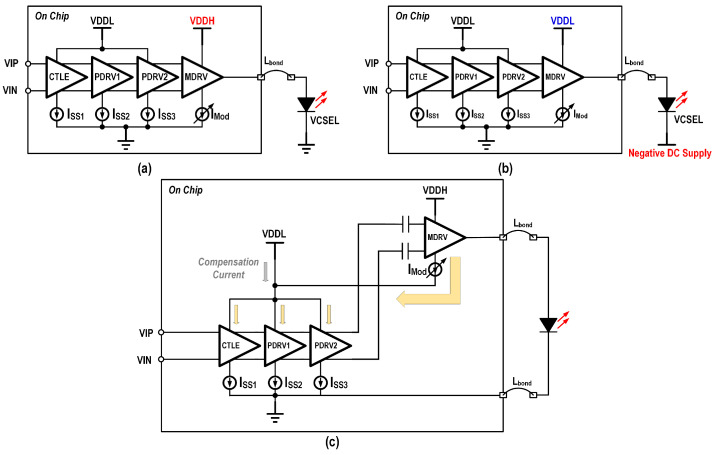
Schematic of the cascaded architecture driver with (**a**) high voltage supply and (**b**) negative supply (**c**) stack architectures.

**Figure 9 micromachines-16-00587-f009:**
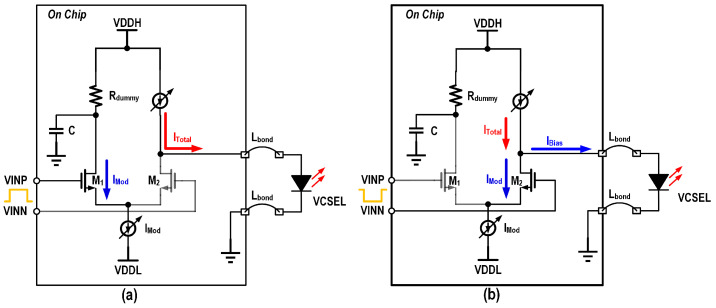
Schematic and operation of the output stage with (**a**) “1” level output and (**b**) “0” level output.

**Figure 10 micromachines-16-00587-f010:**
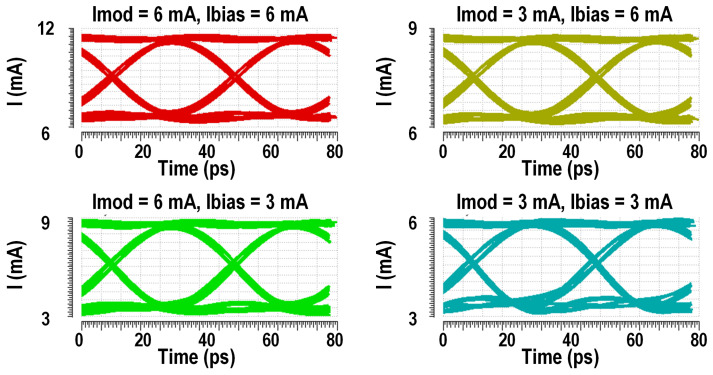
Simulated eye diagrams of the driver.

**Figure 11 micromachines-16-00587-f011:**
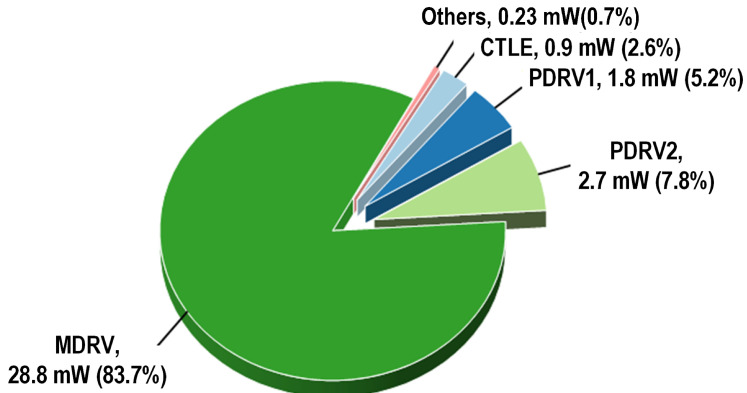
Power breakdown of the driver.

**Figure 12 micromachines-16-00587-f012:**
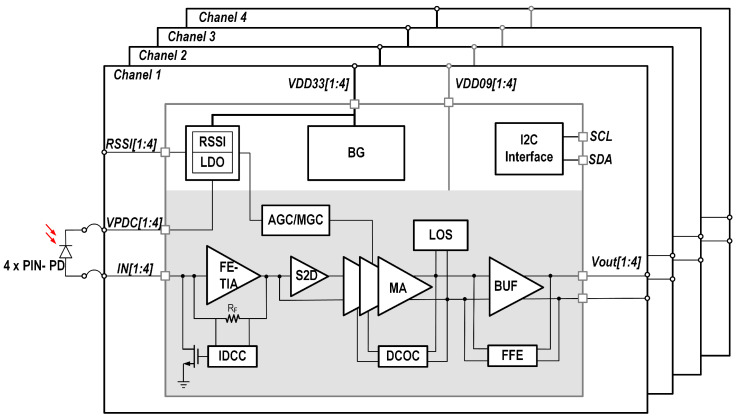
Block diagram of the TIA.

**Figure 13 micromachines-16-00587-f013:**
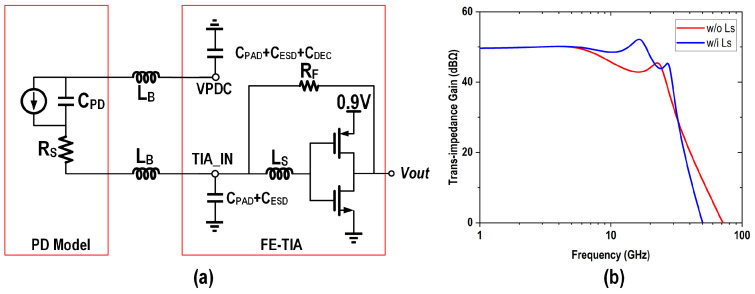
(**a**) Schematic of the FE-TIA with PD and bonding wire model and (**b**) its frequency response.

**Figure 14 micromachines-16-00587-f014:**
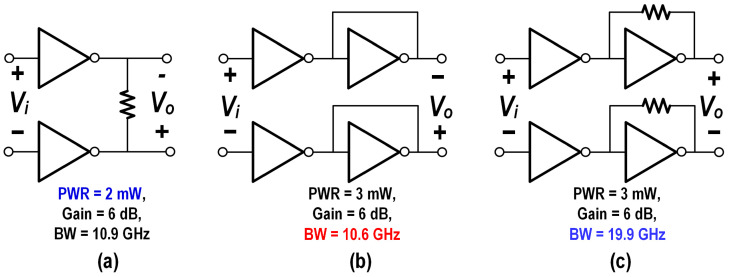
Schematic of inverter-based voltage amplifiers: (**a**) resistive load; (**b**) $g_{m}/g_{m}$; (**c**) Cherry–Hooper structure.

**Figure 15 micromachines-16-00587-f015:**
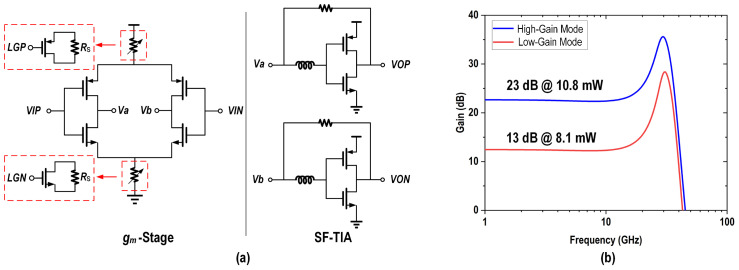
(**a**) Schematic of the tail-current-controlled Cherry–Hooper-based VGA and (**b**) frequency response of the MA.

**Figure 16 micromachines-16-00587-f016:**
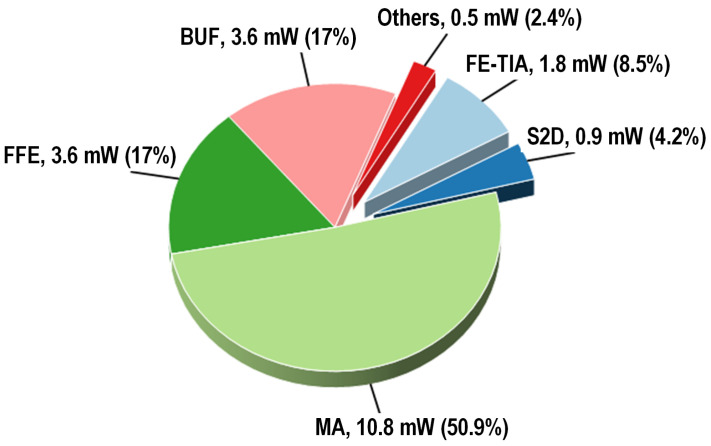
Power breakdown of the TIA.

**Figure 17 micromachines-16-00587-f017:**
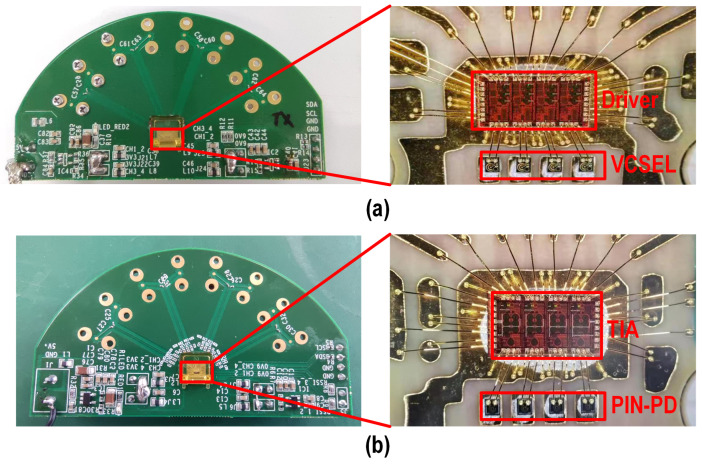
Photograph of the COBs of (**a**) the driver and (**b**) the TIA.

**Figure 18 micromachines-16-00587-f018:**
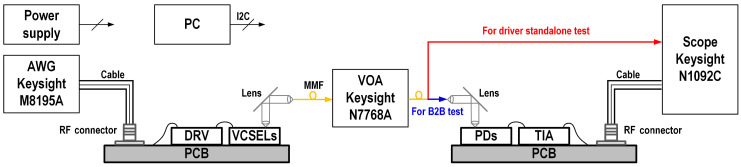
Meaurement setup of the driver and TIA.

**Figure 19 micromachines-16-00587-f019:**
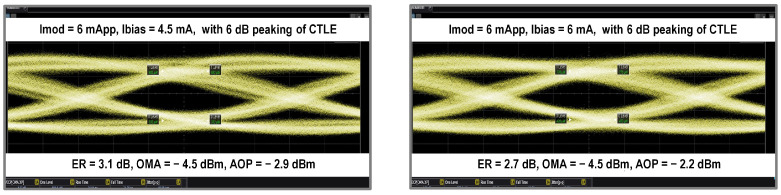
Eye diagrams of the standalone driver test.

**Figure 20 micromachines-16-00587-f020:**
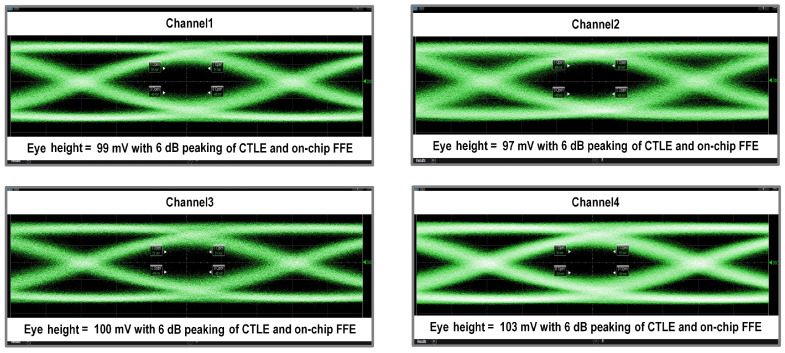
Eye diagrams of the back to back test.

**Figure 21 micromachines-16-00587-f021:**
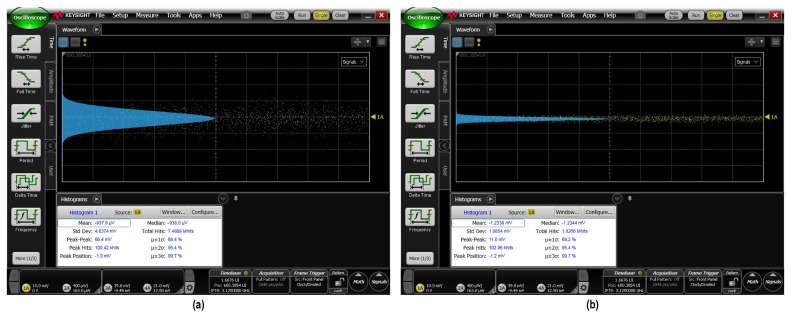
Histogram of (**a**) the measured total output noise and (**b**) the ambient noise.

**Table 1 micromachines-16-00587-t001:** Optical interconnect requirements and power efficiency across scenarios.

Scenarios	Short-Reach	Medium-Reach	Long-Reach
Applications	Intra-rack, LAN	Inter-rack, metro access	Regional networks
Latency	Ultra-low	Moderate	High
Requirements	High density	Scalability	Robustness
Technologies	VCSELs, SiPh, MMF	EMLs, SiPh, SMF	Coherent, EDFAs, WDM
Power Efficiency	1–5 pJ/bit	5–15 pJ/bit	15–100+ pJ/bit

**Table 2 micromachines-16-00587-t002:** Comparison of the structure of optical modules.

Structure	Conv. pluggable	TRO	LPO	CPO
DSP integration	Tx/Rx DSP	Tx DSP	No DSP	No DSP
Power efficiency	10–15 pJ/bit	7–10 pJ/bit	3–5 pJ/bit	1–2 pJ/bit
Thermal	High	Medium	Low	Ultra-low
Reach	10 km+	<2 km	<500 m	<100 m

**Table 3 micromachines-16-00587-t003:** Performance comparison of direct and external modulation devices.

Device	VCSEL	DFB	EML	MZM
Modulation type	Direct	Direct	External	External
Drive swing	<10 mA	∼40 mA	1–2 V	3–5 V
Power consumption	Very low	Low	Moderate	High
Chirp	Moderate	High	Low	Very low
Launch power	Low	Medium	Medium	High
Transmission range	<2 km	∼10 km	∼40 km	>100 km

**Table 4 micromachines-16-00587-t004:** Performance comparisons of drivers.

	[16]	[17]	[18]	This Work
Process	40 nm CMOS	40 nm CMOS	14 nm FinFET	28 nm CMOS
Data rate (Gbps)	25	56	45	25
Signaling	NRZ	PAM4	NRZ	NRZ
Driving type	Anode	Cathode	Anode	Anode
VCSEL BW (GHz)	10	16	20	14
Negative supply	No	No	Yes	No
OMA (dBm)	N/A	−0.9	1.4	−4.5 *
ER (dB)	4.5	N/A	N/A	3.1 *
$I_{\mathrm{mod},\max}$ (${\mathrm{mA}}_{\mathrm{pp}}$)	6	N/A	7	6
Power (mW)	280	97	81.5	29.9 ^†^–39.8 ^‡^
Efficiency (pJ/bit)	11.2	1.73	1.81	1.20 ^†^–1.59 ^‡^

* With VCSEL bias current of 4.5 mA and modulation current of 6 mA; ^†^ with VCSEL bias current of 3 mA; ^‡^ with VCSEL bias current of 6 mA.

**Table 5 micromachines-16-00587-t005:** Performance comparisons of TIAs.

	[19]	[26]	[27]	This Work
Process	65 nm CMOS	65 nm CMOS	40 nm CMOS	28 nm CMOS
Data rate (Gbps)	20	25	30	20
Channel	1	1	1	4
Gain (dBΩ)	78	69.4	63.8	68.4–78.5
IRN (μA)	3.9	3.28	14.9	1.64 ^†^
Power (mW)	45.3	30.8	37.5	13.2 ^†^–21.2 ^‡^
Efficiency (pJ/bit)	1.27	1.23	1.25	0.66 ^†^–1.06 ^‡^

^†^ With low gain, FFE disabled, and 100 mVppd output mode; ^‡^ with high gain, FFE enabled, and 200 mVppd output mode.

## Data Availability

Data are contained within this article.

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
