# Peer review of "CMOS Low-Power Optical Transceiver for Short Reach"

_micromachines, 2025, doi:10.3390/mi16050587_

Round 1
Reviewer 1 Report
Comments and Suggestions for Authors
Ruixuan Yang et al., CMOS Low-Power Optical Transceiver for Short Reach
MDPI, Micrmachines
General Observations
- What is the main question addressed by the research?
Communications between near systems using opto-electronic techniques.
- What parts do you consider original or relevant to the field? What specific gap in the field does the paper address?
The originality of the present research is in employing faster components resulting in an overall faster system.
- What does it add to the subject area compared with other published material?
The authors claim the speed; however, no data is given about competitive methods.
- What specific improvements should the authors consider regarding the methodology?
Improving the writing to make their contribution more clearly recognized by the readers.
- Are the conclusions consistent with the evidence and arguments presented?
The conclusions restated the introductory paragraph, the last one.
- Were all the main questions posed addressed? By which specific experiments?
The authors faied to pose the questions, therefore theere were no questions to be addressed.
- Are the references appropriate?
There are scarecely ny references presented in the Introduction to describe the state of the art prior to this publication. This is the principal shortcoming of this work.
- Comments on the tables and figures and the quality of the data?
Tables and figures are fine. Not much information about the data.
Specific comments
Introduction
References are missing
The objective of the research is not clearly stated. The authors describe their solution to near communicaation issue.
Line 50
»This paper presents the design and implementation of a low power 4×20 Gbps op-50
tical transceiver chipset, fabricated using standard 28-nm CMOS technology. «
Author Response
Comments to the Author
- IntroductionReferences are missing
Authors’ response: Thanks for the careful review and comments.
Action taken:
We added references for the introduction in the new manuscript to clarify the data and the state of the art prior to this publication.
- The objective of the research is not clearly stated. The authors describe their solution to near communication issue.
Authors’ response: Thanks for the careful review and comments.
Action taken:
We have rewritten the last paragraph of the introduction in the new manuscript, adding comparisons with previous state of the art to specify the focus of our work – low power design methodology for short-reach optical transceivers for improved power efficiency.
- Line 50: »This paper presents the design and implementation of a low power 4×20 Gbps optical transceiver chipset, fabricated using standard 28-nm CMOS technology. «
Authors’ response: Thanks for the careful review and comments.
Action taken:
We have rewritten the last paragraph where this sentence was to avoid repetition with the abstract, refined the “low-power design” focus of this work and briefly explained the content and structure of the subsequent article to make it better understanding for the readers.
Reviewer 2 Report
Comments and Suggestions for Authors
Yang, et al. presented a research for the lower-power optical transceiver for short reach. The authors proposed and implemented the new design for the CMOS optical transceiver. By using a tail-current-controlled Cherry-Hooper-based variable gain amplifier, the authors achieved a power efficiency of 1.06 pJ/bit. The research is interesting and timely. I suggest some minor changes before publications.
In the abstract, AI is not the only driving force for the demand of high-bandwidth interconnections. The authors should discuss this in a broader view.
Author Response
Reviewer Comment 1:
Yang, et al. presented a research for the lower-power optical transceiver for short reach. The authors proposed and implemented the new design for the CMOS optical transceiver. By using a tail-current-controlled Cherry-Hooper-based variable gain amplifier, the authors achieved a power efficiency of 1.06 pJ/bit. The research is interesting and timely. I suggest some minor changes before publications.
Author Response: Thank you for your recognition.
Reviewer Comment 2: In the abstract, AI is not the only driving force for the demand of high-bandwidth interconnections. The authors should discuss this in a broader view.
Author Response: We sincerely appreciate the reviewer's insightful suggestion to broaden the discussion of driving forces for high-bandwidth interconnections beyond AI. We have revised the manuscript to address this comment as follows:
The emergence of the AI era driven by Large Language Models (LLMs), and the next-generation high-definition multimedia interface for immersive technologies (AR/VR/metaverse), have created an unprecedented demand for high-bandwidth interconnects.
We believe these changes provide a more comprehensive view of the interconnect landscape while maintaining the technical focus of our work. Thank you for this valuable suggestion.
Reviewer 3 Report
Comments and Suggestions for Authors
Review Micromachines (micromachines-3589908)
Title: “CMOS Low-Power Optical Transceiver for Short Reach”
Authors: Ruixuan Yang, Yiming Dang, Jinhao Chen, Dan Li, Francesco Svelto
Comments:
The work by Yang et al. addresses the power efficiency challenges in CMOS optical transceiver design, leveraging the inherent cost and integration advantages of CMOS technology. The authors present a comprehensive design of a low-power optical transceiver chipset implemented in 28nm CMOS.
The paper is well-presented and well-organized. The manuscript is clearly written, and the simulations, experiments, and analyses are thorough. The authors demonstrate successful data transmission at 4×20 Gbps with an overall power efficiency of 2.65 pJ/bit—an impressive result. This work represents a significant contribution to the progress and development of the field. I recommend the manuscript for publication, pending minor revisions.
Additionally, I have the following minor comment for the authors:
- The authors utilize VCSELs in this work; however, no specific information is provided about them. For example, what is the emission wavelength? Are the VCSELs single-mode or multimode? These details are important, as they affect the choice of detectors and fibers used. Providing this information would help other researchers and readers better understand the system configuration.
Overall, this work is both interesting and impactful. The design and experimental results are impressive. I recommend this manuscript for publication, subject to the minor revision mentioned above.
End
Author Response
Reviewer Comment 1:
The work by Yang et al. addresses the power efficiency challenges in CMOS optical transceiver design, leveraging the inherent cost and integration advantages of CMOS technology. The authors present a comprehensive design of a low-power optical transceiver chipset implemented in 28nm CMOS.
The paper is well-presented and well-organized. The manuscript is clearly written, and the simulations, experiments, and analyses are thorough. The authors demonstrate successful data transmission at 4×20 Gbps with an overall power efficiency of 2.65 pJ/bit—an impressive result. This work represents a significant contribution to the progress and development of the field. I recommend the manuscript for publication, pending minor revisions.
Author Response: Thank you for your recognition.
Reviewer Comment 2: "The authors utilize VCSELs in this work; however, no specific information is provided about them. For example, what is the emission wavelength? Are the VCSELs single-mode or multimode? These details are important, as they affect the choice of detectors and fibers used. Providing this information would help other researchers and readers better understand the system configuration."
Author Response: We agree with the reviewer that the specifications of the VCSELs are critical for reproducibility and system understanding. In the revised manuscript, we have added the following details in Section 4 "Measurement Results": "Both the driver and TIA are directly packaged on board (COB), as shown in Fig. 17, directly wire-bonded with 14-GHz bandwidth 850-nm multimode VCSELs and PIN-PDs respectively."
We believe these revisions enhance the technical rigor and reproducibility of the work. Thank you for highlighting this oversight.